# Recent Advances in Dialogue Machine Translation

**Siyou Liu** [1,†] , **Yuqi Sun** [2] **and Longyue Wang** [3,*]

1   Macao Polytechnic Institute, School of Languages and Translation, Macao, China; violetal@ipm.edu.mo
2   Department of Portuguese, Faculty of Arts and Humanities, University of Macau, Macao, China; sunyuqi@um.edu.mo
3   NLP Centre, AI Lab, Tencent, Shenzhen 518000, China
*   Correspondence: vincentwang0229@gmail.com; Tel.: +86-755-86013388 (ext. 57508)
†   Current address: B210, Macao Polytechnic Institute, Rua de Luís Gonzaga Gomes, Macao, China.

**Abstract:** Recent years have seen a surge of interest in dialogue translation, which is a significant application task for machine translation (MT) technology. However, this has so far not been extensively explored due to its inherent characteristics including data limitation, discourse properties and personality traits. In this article, we give the first comprehensive review of dialogue MT, including well-defined problems (e.g., 4 perspectives), collected resources (e.g., 5 language pairs and 4 sub-domains), representative approaches (e.g., architecture, discourse phenomena and personality) and useful applications (e.g., hotel-booking chat system). After systematical investigation, we also build a state-of-the-art dialogue NMT system by leveraging a breadth of established approaches such as novel architectures, popular pre-training and advanced techniques. Encouragingly, we push the state-of-the-art performance up to 62.7 BLEU points on a commonly-used benchmark by using mBART pre-training. We hope that this survey paper could significantly promote the research in dialogue MT.

**Keywords:** dialogue; neural machine translation; discourse issue; benchmark data; existing approaches; real-life applications; building advanced system

## 1. Introduction

Dialogue is a written or spoken conversational exchange between two or more people, expressing human emotions, moods, attitudes, and personality [1]. Nowadays there is a huge demand for cross-language dialogue communication between people, and advances in machine translation (MT) for improving communication (or translation) have been seen in recent years. MT is a sequence-to-sequence prediction task, which aims to find for a source language sentence the most probable target language sentence that shares the most similar meaning. Dialogue machine translation comprises a number of significant application domains such as audiovisual subtitles, meeting transcripts, instant messaging, and speech-to-speech interpretation. Although neural machine translation (NMT) [2–4] has achieved great progress in recent years, translating dialogues is still a challenging task due to its inherent characteristics such as data limitation, irregular expressions, discourse properties, and personality traits. To address corresponding problems, a number of works are exploited to improve the translation quality of dialogue translation systems.

Although some researchers have explored ways to construct data for modeling dialogues [5–9], parallel data are still scarce to build robust dialogue translation models. As a result, previous work has been hampered by a lack of dialogue-domain datasets [10–12]. In contrast to the translation of general domains (e.g., news), in which the text is carefully authored and well formatted, translating dialogue conversations has been less planned, more informal, and often discourse-aware. One research direction investigates incorporating dialogue history into document-level NMT architectures [13,14], which aims to implicitly enhance the ability on modeling coherence and consistency. On the other hand,

some research has explicitly modelled various discourse phenomena in dialogues, such as anaphora [15–17] and discourse connectives [18,19]. Furthermore, recent studies investigated effects of inherent characteristics on translating dialogues, including speaker information [20], role preference [21] and topics [11].

In recent years, there have been more interest in modeling dialogue machine translation. In this article, we aim to give a comprehensive survey of the recent advances in dialogue MT. First of all, we systematically define four critical problems in dialogue translation by reviewing a large number of related works. Second, we collect nearly all existing corpora for the dialogue translation task, covering 5 language pairs and 4 sub-domains. Third, we also respectively introduce three representative approaches on architecture, discourse phenomenon and personality aspects. Last, we discuss an example of real-life applications, demonstrating the importance and feasibility of a dialogue translation system. Furthermore, we explore the potential of building a state-of-the-art dialogue translation system by leveraging a breadth of established approaches. Experiments are conducted on a task-oriented translation dataset that is widely used in previous studies (i.e., WMT20 English-German). Encouragingly, we push the SOTA performance up to 62.7 BLEU points on the benchmark by using the mBART pre-training method.

This paper describes highlights of recent advances in dialogue machine translation:

1.  Previous works mainly exploited dialogue MT from perspectives of coherence, consistency, and cohesion. Furthermore, recent studies began to pay more attention to the issue of personality such as role preference.
2.  Although there are some related corpora, the scarcity of training data remains one of the crucial issues, which severely hinders the further development of the deep learning methods for real applications of dialogue translation.
3.  Existing approaches can be categorized into three main strands. One research line is to exploit document-level NMT architectures, which can improve the consistency and coherence in translation output. The second one tries to deal with specific discourse phenomena such as anaphora, which can lead to better cohesion in translations. The third line aims to enhance the personality of dialogue MT systems by leveraging additional information labeled by humans. In future work, it is necessary to design an end-to-end model that can capture various characteristics of dialogues.
4.  Through our empirical experiments, we gain some interesting findings: (1) data selection methods can significantly improve the baseline model especially for small-scale data; (2) the large batch learning works well, which makes sentence-level NMT models perform the best among different NMT models; (3) document-level contexts are not always useful on the dialogue translation due to the limitation of data; (4) it is helpful to dialogue MT by transferring general knowledge from pretrained models.

This section is organized as follows: we first introduce the fundamental knowledge on NMT (including the models, frameworks, and evaluation metrics) and basic information on dialogue translation (including theory, definition and characteristics) in Section 2. Section 3 gives a comprehensive review of problems, resources, approaches, and real-life applications for dialogue translation task. We explore building a state-of-the-art dialogue translation system by combining advanced techniques in Section 4. Finally, we summarize the content of this article in Section 5.

## 2. Preliminary

Without loss of generality, we provide the fundamental knowledge on machine translation and dialogue translation in this section.

### 2.1. Machine Translation

As an active research field in NLP, the task of MT is to translate texts from one language to another language. It is a challenging task for MT to generate high-quality translation, because computers need to thoroughly understand the text in the source language and have a good knowledge of the target language. In the last several decades, scientific research

in the field of MT has experienced three main historical periods including Rule-based Machine Translation (RBMT) [22], Statistical Machine Translation (SMT) [23] and Neural Machine Translation (NMT) [24,25], and each of these models has significantly improved the performance of MT systems.

### 2.1.1. Statistical Machine Translation

Assume that a sentence pair $\mathbf{x} = \{x_1, \ldots, x_i, \ldots, x_I\}$ and $\mathbf{y} = \{y_1, \ldots, y_j, \ldots, y_J\}$ are in source and target side, respectively. $x_i$ is the $i$-th word of $\mathbf{x}$ and $y_j$ is the $j$-th word of $\mathbf{y}$. $I$ and $J$ are lengths of $\mathbf{x}$ and $\mathbf{y}$, which can be different. Based on Bayes decision theory, we can formulate SMT [26] as:

$$\hat{\mathbf{y}} = \arg\max_{\mathbf{y}} p(\mathbf{y}|\mathbf{x}) \propto \arg\max_{\mathbf{y}} p(\mathbf{x}|\mathbf{y})p(\mathbf{y}) \tag{1}$$

where $\hat{\mathbf{y}}$ denotes the translation output with the highest translation probability. The translation problem is factored into $p(\mathbf{x}|\mathbf{y})$ and $p(\mathbf{y})$, representing the inverse translation probability and language model probability respectively. The denominator $p(\mathbf{x})$ is ignored since it remains constant for a given source sentence $\mathbf{x}$. The advantage of this decomposition is that we can learn separate probabilities in order to compute $\hat{\mathbf{y}}$. Och and Ney [27] proposed a log-linear model, which incorporates different features containing information from the source and target sentences in the model, in addition to the language and translation models of the original noisy channel [28] approach. Figure 1a describes the architectures of phrase-based SMT [29], which consists of several components: (1) words within the parallel corpus are aligned and phrase pairs are then extracted based on word-alignment results [30]; (2) the translation model and the lexicalized reordering model can be learned using aligned phrases; (3) an $n$-gram language model can be built using a large number of monolingual sentences in the target language [31]; (4) these models are optimized under the log-linear framework in order to maximize the performance using a development set [32]; (5) with the optimized weight parameters of the features in the models, we can finally translate the test set and the evaluation score indicates the performance of the whole system.

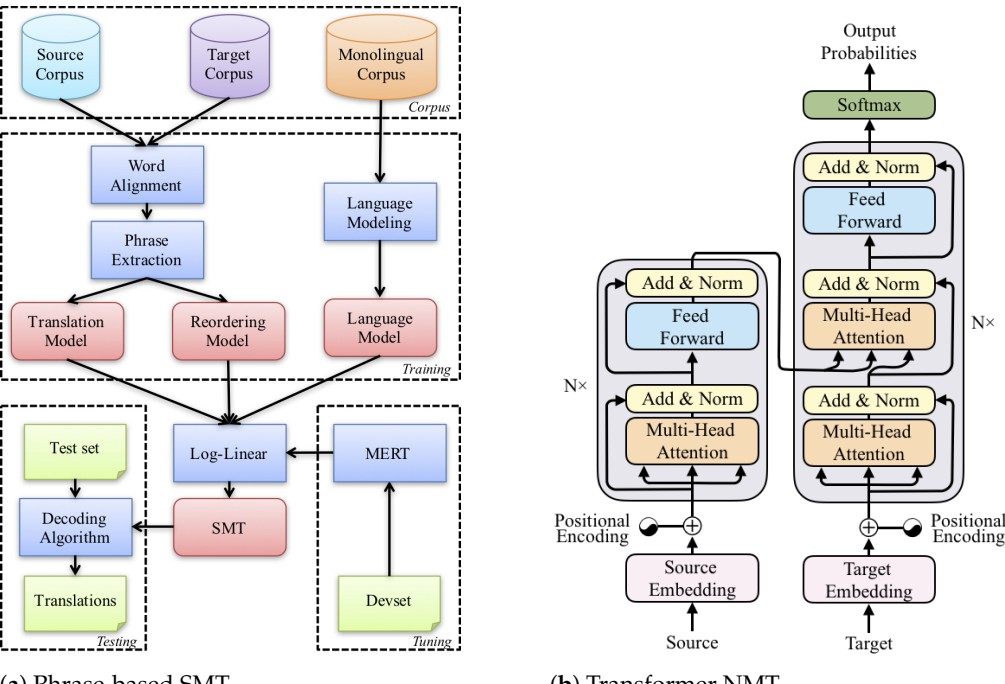

(**a**) Phrase-based SMT  (**b**) Transformer NMT

**Figure 1.** Architectures of the (**a**) SMT and (**b**) NMT models.

### 2.1.2. Neural Machine Translation

In recent years, NMT [2,24,25] has made significant progress towards constructing and utilizing a single large neural network to handle the entire translation task. A standard NMT model directly optimizes the conditional probability of a target sentence $\mathbf{y} = y_1, \ldots, y_J$ given its corresponding source sentence $\mathbf{x} = x_1, \ldots, x_I$:

$$P(\mathbf{y}|\mathbf{x}; \theta) = \prod_{j=1}^{J} P(y_j|\mathbf{y}_{<j}, \mathbf{x}; \theta) \tag{2}$$

where $\theta$ is a set of model parameters and $\mathbf{y}_{<j}$ denotes the partial translation. The probability $P(\mathbf{y}|\mathbf{x}; \theta)$ is defined on the neural network based encoder-decoder framework [25,33], where the encoder summarizes the source sentence into a sequence of representations $\mathbf{H} = \mathbf{H}_1, \ldots, \mathbf{H}_I$ with $\mathbf{H} \in \mathbb{R}^{I \times d}$, and the decoder generates target words based on the representations. Typically, this framework can be implemented as a recurrent neural network (RNN) [2], convolutional neural network (CNN) [4], and Transformer [3]. The Transformer has emerged as the dominant NMT paradigm among the different models, as shown in Figure 1.

The parameters of the NMT model are trained to maximize the likelihood of a set of training examples $D = \{[\mathbf{x}^m, \mathbf{y}^m]\}_{m=1}^{M}$:

$$\mathcal{L}(\theta) = \arg\max_{\theta} \sum_{m=1}^{M} \log P(\mathbf{y}^m|\mathbf{x}^m; \theta) \tag{3}$$

which is used as a sentence-level baseline in this work.

We use automatic evaluation metrics to evaluate the translation quality. BLEU [34] is the most commonly-used one, which is reference-based and computed over the entire test set. The output of BLEU is a score between 0 and 100%, indicating the similarity between the MT outputs and the reference translations. The higher the score is, the better the translation is. It is computed based on a modified *n*-gram precision:

$$BLEU = BP * \exp \sum_{n=1}^{N} \frac{1}{N} \log(\frac{|m_n \cap m_r|}{|m_n|}) \tag{4}$$

where *n* represents the order of the *n*-grams compared between the translations and references. Typically, *n* is from 1 to 4. $m_n$ and $m_r$ indicate the *n*-grams occurring in the MT outputs and the corresponding references respectively. $|m_n \cap m_r|$ is the number of *n*-grams occurring in both translations and references. *BP* is the brevity penalty to penalize shorter translations than the references.

### 2.2. Dialogue Translation

Dialogue is a written or spoken conversational exchange between two or more people, and a literary and theatrical form that depicts such an exchange. It is an essential component of social behaviour to express human emotions, moods, attitudes, and personality. In the context of dialogue modeling, we divided the dialogue into two types: task-oriented and open-domain. Specifically, the task-oriented dialogue system makes users communicate in a task-based fashion: (1) help users achieve their specific goals; (2) focus on understanding users, tracking states, and generating subsequent actions; (3) minimize the number of turns (i.e., fewer turns the better). On the other hand, an open-domain dialogue system aims to establish long-term connections with users by satisfying the human need for communication, affection, and social belonging.

A typical scenario for such application is translating dialogue texts, in particular the record of group chats or movie subtitles, which helps people of different languages understand cross-language chat and improve their comprehension capabilities. For instance,

Figures 2 and 3 show examples of open-domain and task-oriented dialogue translation scenarios, respectively.

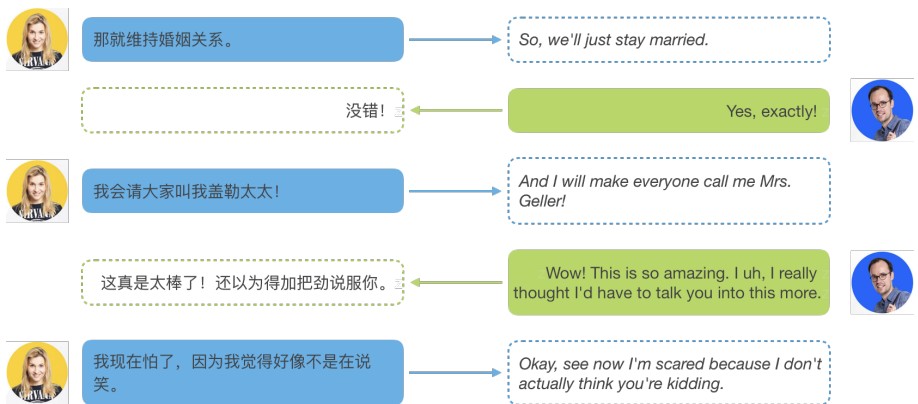

**Figure 2.** An example of chat-based dialogue translation. It is extracted from the Chinese-English MVSub Corpus.

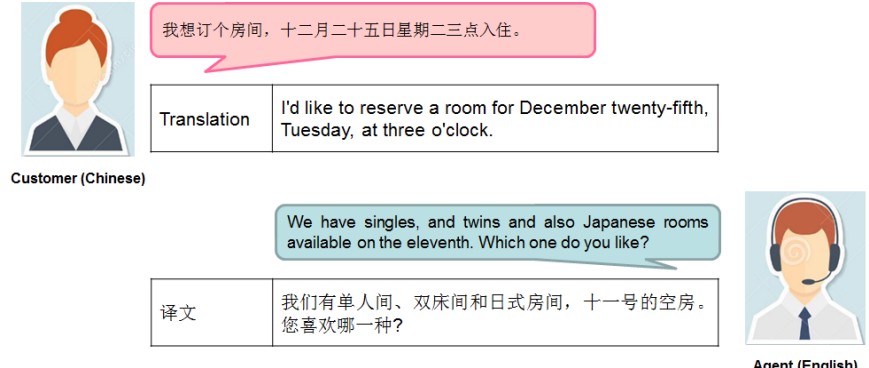

**Figure 3.** An example of task-based dialogue translation. It is extracted from the Chinese-English IWSLT-DIALOG Corpus.

Although NMT has achieved great progress in recent years, translating conversational text is still an important and challenging application task. In contrast to the translation of common domains (e.g., newswire and biomedical), in which the text is carefully authored and well-formatted, translating dialogue conversations is less planned, more informal, and often context-aware. More specifically, few researchers have investigated how to improve the MT of conversational material by exploiting their internal structure. This lack of research on the dialogue MT is a surprising fact, since dialogue exhibits more cohesiveness than a single sentence and at least as much as textual discourse. In natural dialogues, speakers may make some kinds of mistakes or so called irregular expressions. One of the most challenging problems which dialogue MT must deal with is translating irregular expressions in the natural conversation, such as ungrammatical, incompleted, or ill-formed sentences. However, most existing machine translation systems reject utterances with irregular expressions. Furthermore, this task has so far not been extensively explored largely due to the lack of publicly available datasets.

## 3. Overview of Dialogue Machine Translation

In this section, we make a survey of problems (in Section 3.1), resources (in Section 3.2), approaches (in Section 3.3), and real-life applications (in Section 3.4) for a dialogue machine translation task.

### 3.1. Dialogue Translation Issues

Dialogue machine translation varies from the other translation tasks, e.g., news and biomedical, mainly due to the fact that the conversations are bilingual, less planned, more informal, and often discourse-aware. Furthermore, such conversations are usually characterized by shorter and simpler sentences and contain more implicit information. According to the inherent characteristics of dialogue, we divide the issues of dialogue translation into four perspectives: coherence, consistency, cohesion, and personality. As shown in Figure 4, each perspective contains its sub-fields and related works. Note that most methods are used in general-domain translation, but can also be employed for dialogue translation task.

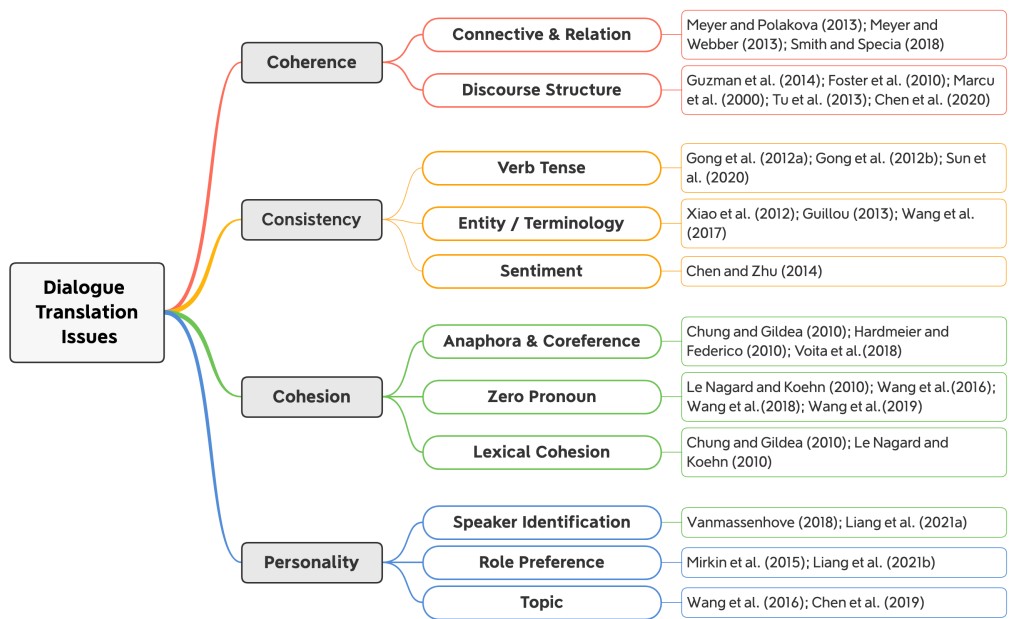

**Figure 4.** The overview of dialogue translation issues and sub-categories along with related works.

**Coherence** is created referentially when different parts of a text refer to the same entities, and relationally, by means of coherence relations such as "Cause–Consequence" between different discourse segments [35]. Some researchers attempt to exploit the discourse trees (e.g., Rhetorical Structure Theory [36]) of the input texts to infer more coherent translations [37–41]. Another research line investigates effects of specific phenomena, such as discourse connectives and relations on MT [18,19,42]. Besides, document-level NMT architectures are proposed to implicitly modeling information across sentences [13,14].

**Consistency** is another critical issue in dialogue MT, where a repeated term should keep the same translation throughout the whole text [43]. The underlying assumption is that the same concepts should be consistently referred to with the same words in a translation. To alleviate the inconsistency problems, some researchers have investigated different approaches for MT and evaluation, which can be divided into different aspects, such as verb tense [44–46], entity/terminology [43,47], sentiment [48]. Furthermore, document-level NMT can also improve translation consistency, including cache-based [49,50], and document-level decoding [51,52] and document-level architecture [13,53].

**Cohesion** is a surface property of the text that is realized by explicit clues. It occurs whenever "the interpretation of some element in the discourse is dependent on that of another" [54]. Some researchers have investigated approaches of incorporating anaphora/coreference information to improve the performance of MT [51,55]. Zero pronoun (ZP) is a more complex case of anaphora, where pronouns are often omitted when they can be pragmatically or grammatically inferable from intra- and inter-sentential contexts [56]. This severely harms MT systems since the translation of such missing pronouns cannot be normally reproduced, and several works have addressed this prob-

lem [15,16,57,58]. Lexical cohesion refers to the way related words are chosen to link elements of a text. Some studies have tried to model lexical cohesion for both MT and evaluation tasks [59,60].

**Personality** is the specific set of qualities and interests that make a person unique and unlike others. It is one of the major challenges in conversational systems, which aims to present a consistent personality [61]. Due to the lack of explicitly modeling such inherent characteristics (e.g., role preference), dialogue translation systems cannot obtain satisfactory results [12]. Therefore, recent studies have investigated different inherent characteristics of dialogue translation, including speaker identification [20,62], role preference [21,63], and topic [11,64].

### 3.2. Existing Data

Translating dialogue has so far not been extensively explored in prior MT research, largely due to the lack of publicly available data sets [12]. Prior related work has mostly focused on movie subtitles and European Parliament speeches. To alleviate this problem, the WMT2020 Shared Task (https://www.statmt.org/wmt20, accessed on 20 November 2021) created a corpus on task-oriented dialogue translation, namely BConTrasT [9].

Some work regarding bilingual subtitles as parallel corpora exists, but it lacks rich information between utterances [10,65–70]. Other work focuses on mining the internal structure in dialogue data from movie scripts. However, these are monolingual data, which cannot be used for MT [5–8]. In general, the fact is that bilingual subtitles are ideal resources to extract parallel sentence-level utterances, and movie scripts contain rich information such as dialogue boundaries and speaker tags. Recently, some works explored constructing parallel dialogue data with rich information [11,15,20].

The detailed corpora for the dialogue translation task are summarized as follows and in Table 1.

**Table 1.** Statistics of training corpora for dialogue machine translation. The details are the name of corpora, language pairs, domains, number of sentences ($|S|$), number of documents ($|D|$), averaged sentence length ($|L|$). K stands for thousand and M for million.

| Corpus | Language | Domain | $|S|$ | $|D|$ | $|L|$ |
|---|---|---|---|---|---|
| OpenSubtitle | FR-EN | movie subtitle | 29.2 M | 35 K | 8.0/7.5 |
| | ES-EN | | 64.7 M | 78 K | 8.0/7.3 |
| | EN-RU | | 27.4 M | 35 K | 5.8/6.7 |
| | ZH-EN | | 11.2 M | 14 K | 5.4/7.3 |
| TVSub | ZH-EN | TV series subtitle | 2.2 M | 3 K | 5.6/7.7 |
| MVSub | ZH-EN | *Friends* subtitle | 0.1 M | 5 K | 6.0/7.9 |
| IWSLT-DIALOG | ZH-EN | travel dialogue | 0.2 M | 2 K | 19.5/21.0 |
| BConTrasT | EN-DE | task-based dialogue | 8.1 K | 0.6 K | 6.7/9.2 |
| BMELD | ZH-EN | *Friends* subtitle | 6.2 K | 1 K | 6.0/7.9 |
| Europarl | ET-EN | European Parliament speech | 0.2 M | 150 K | 35.1/36.4 |
| | EN-DE | | 1.9 M | - | 23.2/24.9 |
| | FR-EN | | 2.0 M | - | 25.6/25.0 |

**OpenSubtitle** (http://opus.nlpl.eu/OpenSubtitles2018.php, accessed on 20 November 2021) is a collection of translated movie subtitles [71], which are originally crawled from the movie subtitle website (http://www.opensubtitles.org, accessed on 20 November 2021). Bilingual subtitles are ideal resources to extract parallel utterances because a large amount of data are available. Most of the translations of subtitles are usually simple and short, and they do not preserve the syntactic structures of their original sentences at all. Previous works on dialogue translation usually randomly select some episodes as the validation set, and the others as the test set. In total, it contains 62 language pairs, and researchers mainly exploited commonly-cited French–English, Spanish–English and Russian–English.

**TVSub** (https://github.com/longyuewangdcu/tvsub, accessed on 20 November 2021) extracted subtitles from TV episodes, instead of movies compared with the OpenSubtitle Corpus [15]. The dataset is the Chinese–English language pair. Its source-side sentences are automatically annotated with zero pronouns by a heuristic algorithm [58] (The annotation indicates recovering dropped pronouns with correct pronoun words). Thus, it can be generally used to study dialogue translation as well as the zero anaphora phenomenon. More than two million sentence pairs were extracted from the subtitles of television episodes. Their multiple references and zero pronoun labels in validation and test sets have been manually designed.

**MVSub** (http://longyuewang.com/corpora/resource.html, accessed on 20 November 2021) is extracted from a classic American TV series, namely Friends [11]. It contains speaker tags and scene boundaries, which are all manually annotated according to their corresponding screenplay scripts. Thus, it can be generally used to study dialogue translation as well as personality characteristics. The dataset contains 100 thousand Chinese–English sentence pairs, and validation and test sets are well designed.

**IWSLT-DIALOG** (http://iwslt2010.fbk.eu/node/33, accessed on 20 November 2021) are from the Spoken Language Databases (SLDB) corpus, a collection of human-mediated cross-lingual dialogues in travel situations. In addition, parts of the BTEC corpus are also provided to the participants of the DIALOG Task [72]. The dataset contains very limited Chinese–English sentence pairs. The validation and test sets are not available. Thus researchers usually randomly selected parts of data. Ref. [73] pointed out that NMT systems have a steeper learning curve with respect to the amount of training data, resulting in worse quality in low-resource settings. The DIALOG is difficult to translate given the variety of topics in quite small-scale training data.

**BConTrasT** (https://github.com/Unbabel/BConTrasT, accessed on 20 November 2021) is first provided by WMT 2020 Chat Translation Task, which is translated from English into German and is based on the monolingual Taskmaster-1 corpus [9]. The conversations (originally in English) were first automatically translated into German and then manually post-edited by human editors, who are native German speakers. Having the conversations in both languages allows us to simulate bilingual conversations in which one speaker, the customer, speaks in German and the other speaker, the agent, answers in English. The training, validation and test sets contain utterances in task-based dialogues with contextual information.

**BMELD** (https://github.com/XL2248/CPCC, accessed on 20 November 2021) is created based on the dialogue dataset in the MELD (originally in English) [74]. Ref. [20] firstly crawled the corresponding Chinese translations from movie website and then manually post-edited them according to the dialogue history by native Chinese speakers, who are postgraduate students majoring in English. Finally, they assume 50% of speakers as Chinese to keep data balance for Chinese-to-English translations and build the bilingual MELD (BMELD). The MELD is a multi-modal emotionLines dialogue dataset, each utterance of which corresponds to a video, voice, and text, and is annotated with emotion and sentiment.

**Europarl** (https://www.statmt.org/europarl, accessed on 20 November 2021) is extracted from the proceedings of the European Parliament. Sentences are usually long and formally used in the official conference. It contains 21 European language pairs [75].

*3.3. Representative Approaches*

As discussed in Section 3.1, there are different strands of research in the literature. One attempts to exploit the macroscopic structure of the input texts to infer better translations in terms of discourse properties, including cohesion, coherence, and consistency. Other work deals with specific linguistic phenomena that are governed by discourse-level processes, such as the generation of anaphoric pronouns and translation of discourse connectives. These strands are not isolated, but closely related to each other. For instance, document-level information can not only improve the overall performance of MT but also alleviate

inconsistency problems at the same time. Furthermore, some researchers investigated effects of characteristics of dialogue on MT [11,20,21]. Instead of reviewing all existing approaches, we mainly introduce three representative ones: document-level architecture, discourse phenomena for dialogue MT, and translation with speaker information.

### 3.3.1. Architecture: Document-Level NMT

It aims to consider both the current sentence and its large context in a unified model to improve translation performances, especially discourse properties. Figure 5 introduces a classic document-level NMT model, namely *multi-encoder* [13,53,55]. Taking [55] for an example, it employs $(N-1)\times$ layers of context encoder to summarize the larger context from source-side previous sentences, and $(N-1)\times$ layers of a standard encoder to model the current sentence. At the last layer, they integrate the contextual information with the source representations using a gating mechanism. Finally, the combined document-level representations are fed into the NMT decoder to translate the current sentence.

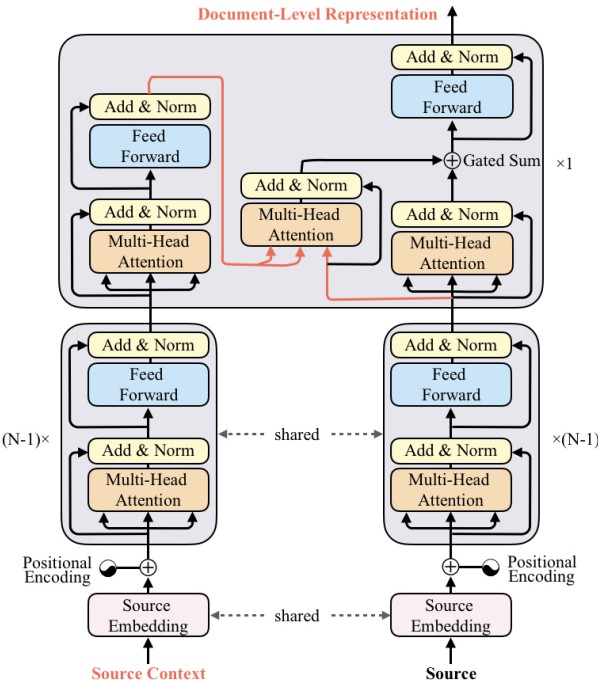

**Figure 5.** The architecture of the document-level Transformer model.

Given a source sentence $\mathbf{x}_i$ to be translated, we can consider its $K$ previous sentences in the same document as source context $C = \{\mathbf{x}_{i-K}, \ldots, \mathbf{x}_{i-1}\}$. The source encoder employs multi-head self-attention $\text{ATT}(\cdot)$ to transform an input sentence $\mathbf{x}_i$ into a sequence of representations $\mathbf{O}^h = \{\mathbf{o}_1^h, \ldots, \mathbf{o}_I^h\}$ by:

$$\mathbf{o}_i^h = \text{ATT}(\mathbf{q}_i^h, \mathbf{K}^h)\mathbf{V}^h \in \mathbb{R}^{\frac{d}{H}} \tag{5}$$

where $h$ is one of $H$ heads. $\mathbf{Q}$, $\mathbf{K}$ and $\mathbf{V}$, respectively, represent queries, keys and values, which are calculated as:

$$\mathbf{Q}, \mathbf{K}, \mathbf{V} = \mathbf{X}\mathbf{W}_Q, \mathbf{X}\mathbf{W}_K, \mathbf{X}\mathbf{W}_V \in \mathbb{R}^{I \times d} \tag{6}$$

where $\{\mathbf{W}_Q, \mathbf{W}_K, \mathbf{W}_V\} \in \mathbb{R}^{d \times d}$ are trainable parameters and $d$ indicates the hidden size. The context encoder employs the same networks as the source encoder to obtain the context output $\hat{\mathbf{O}}$. Finally, the two encoder outputs $\mathbf{O}$ and $\hat{\mathbf{O}}$ are combined via a gated sum, as in:

$$
\begin{aligned}
\lambda_d &= \sigma(W_\lambda[O_d, \hat{O}_d] + b_d) && (7) \\
O' &= \lambda_d \odot O_d + (1 - \lambda_d) \odot \hat{O}_d && (8)
\end{aligned}
$$

in which $\sigma(\cdot)$ is the logistic sigmoid function and $W_\lambda$ is the parameter. $O'$ is the final document-level representation, which is further fed into the NMT decoder. Following [55], people usually share the parameters of context encoders and embedding with those of the standard NMT encoder.

### 3.3.2. Discourse Phenomenon: Zero Pronoun Translation

Pronouns are frequently omitted in pro-drop languages (e.g., Chinese and Japanese), generally leading to significant challenges with respect to the production of complete translations. This problem is especially severe in informal genres, such as dialogues and conversation, where pronouns are more frequently omitted to make utterances more compact [76]. Ref. [58] proposed an automatic method to annotate ZPs by utilizing the parallel corpus of MT. The homologous data for both ZP prediction and translation leads to significant improvements in translation performances for both statistical [58] and neural MT models [15]. However, such approaches still require external ZP prediction models with a low accuracy of 66%. The numerous errors of ZP prediction errors will be propagated to translation models, which leads to new translation problems. Therefore, some works began to investigate an end-to-end ZP translation model [15,16].

Taking reconstructor-based NMT [15] for example, the reconstructor reads a sequence of hidden states and the annotated source sentence, and outputs a reconstruction score. It employs an attention model to reconstruct the annotated source sentence $\hat{\mathbf{x}} = \{\hat{x}_1, \hat{x}_2, \ldots, \hat{x}_{J'}\}$ word by word, which is conditioned on the input latent representations $\mathbf{v} = \{\mathbf{v}_1, \mathbf{v}_2, \ldots, \mathbf{v}_T\}$. The reconstruction score is computed by Equation (9):

$$
R(\hat{\mathbf{x}}|\mathbf{v}) = \prod_{j=1}^{J'} R(\hat{x}_j|\hat{x}_{<j}, \mathbf{v}) = \prod_{j=1}^{J'} g_r(\hat{x}_{j-1}, \hat{\mathbf{s}}_j, \hat{\mathbf{c}}_j) \tag{9}
$$

where $\hat{\mathbf{s}}_j$ is the hidden state in the reconstructor, and computed by Equation (10):

$$
\hat{\mathbf{s}}_j = f_r(\hat{x}_{j-1}, \hat{\mathbf{s}}_{j-1}, \hat{\mathbf{c}}_j) \tag{10}
$$

Here, $g_r(\cdot)$ and $f_r(\cdot)$ are, respectively, softmax and activation functions for the reconstructor. The context vector $\hat{\mathbf{c}}_j$ is computed as a weighted sum of hidden states $\mathbf{v}$, as in Equation (11):

$$
\hat{\mathbf{c}}_j = \sum_{t=1}^{T} \hat{\alpha}_{j,t} \cdot \mathbf{v}_t \tag{11}
$$

where the weight $\hat{\alpha}_{j,t}$ is calculated by an additional attention model. The parameters related to the attention model, $g_r(\cdot)$, and $f_r(\cdot)$, are independent of the standard NMT model. The labeled source words $\hat{\mathbf{x}}$ share the same word embeddings with the NMT encoder. Finally, they augment the standard encoder–decoder-based NMT model with the introduced reconstructor, as shown in Figure 6. The standard encoder–decoder reads the source sentence $\mathbf{x}$ and outputs its translation $\mathbf{y}$ along with the likelihood score.

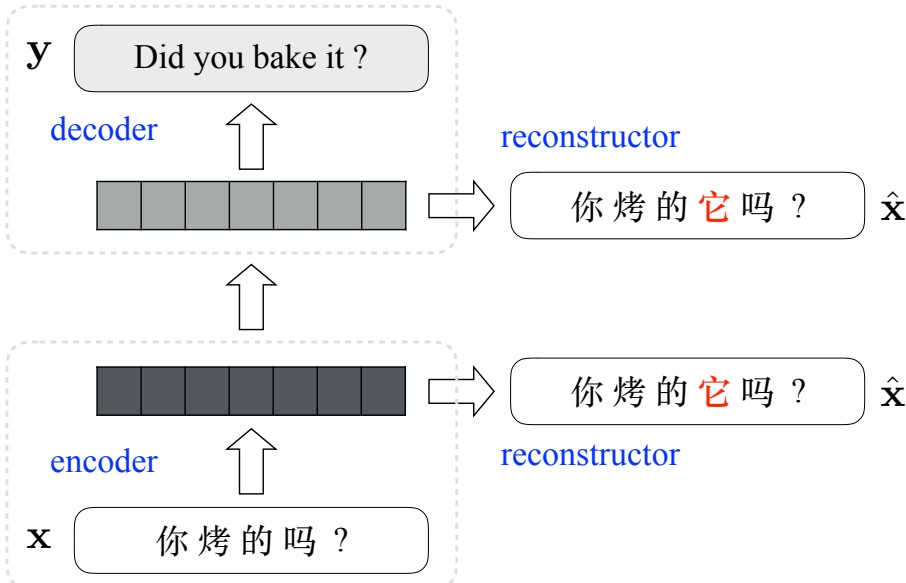

**Figure 6.** The architecture of reconstructor-augmented NMT. The Chinese word in red is a zero pronoun *it*.

We train both the encoder–decoder and the introduced reconstructors together in a single end-to-end process. The training objective can be revised as in Equation (12):

$$
\begin{aligned}
J(\theta,\gamma,\psi) \quad = \quad & \underset{\theta,\gamma,\psi}{\arg\max} \sum_{n=1}^{N} \Bigg\{ \underbrace{\log P(\mathbf{y}^n|\mathbf{x}^n;\theta)}_{likelihood} \\
+ \quad & \lambda \underbrace{\log R_{enc}(\hat{\mathbf{x}}^n|\mathbf{h}^n;\theta,\gamma)}_{enc\text{-}rec} \\
+ \quad & \eta \underbrace{\log R_{dec}(\hat{\mathbf{x}}^n|\mathbf{s}^n;\theta,\psi)}_{dec\text{-}rec} \Bigg\}
\end{aligned}
\tag{12}
$$

where $\theta$ is the parameter matrix in the encoder–decoder, and $\gamma$ and $\psi$ are model parameters related to the *encoder-side reconstructor* ("enc-dec") and *decoder-side reconstructor* ("dec-rec"), respectively. $\lambda$ and $\eta$ are hyper-parameters that balance the preference between likelihood and reconstruction scores; $\mathbf{h}$ and $\mathbf{s}$ are encoder and decoder hidden states. The original training objective $P(\cdot)$ guides the standard NMT counterpart to provide better translations. Furthermore, the auxiliary reconstruction objectives ($R_{enc}(\cdot)$ and $R_{dec}(\cdot)$) guide the related part of the parameter matrix $\theta$ to learn better latent representations, which are used to reconstruct the annotated source sentence. The parameters of the model are trained to maximize the likelihood and reconstruction scores of a set of training examples $\{[\mathbf{x}^n,\mathbf{y}^n]\}_{n=1}^{N}$.

In testing, reconstruction can serve as a re-ranking technique to select a better translation from the *k*-best candidates generated by the decoder. Each translation candidate is assigned a likelihood score from the standard encoder–decoder, as well as reconstruction score(s) from the newly added reconstructor(s). As shown in Figure 7, given an input sentence, a two-phase scheme is used.

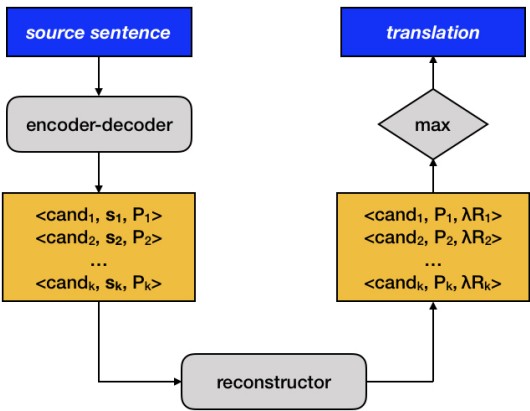

**Figure 7.** Illustration of decoding with reconstruction.

### 3.3.3. Dialogue Personality: Speaker Information

As shown in Figure 8, Ref. [11] conduct a personalized MT experiment to explore the effects of speaker tags on dialogue MT. They first build a baseline MT engine using Moses [29] on a dataset extracted from the bilingual movie subtitle of *Friends*. They train a 5-gram language model using the SRI Language Toolkit [31] on the target side of the parallel corpus. Besides, they use GIZA++ [30] for word alignment and minimum error rate training [32] to optimize feature weights. Based on the hypothesis that different types of speakers may have specific speaking styles, they employ a language model adaptation method to boost the MT system. Instead of building a LM on the whole data, they split the data into two separate parts according to the speakers' sex and then build two separate LMs. As Moses supports multiple LM integration, they directly feed Moses two LMs.

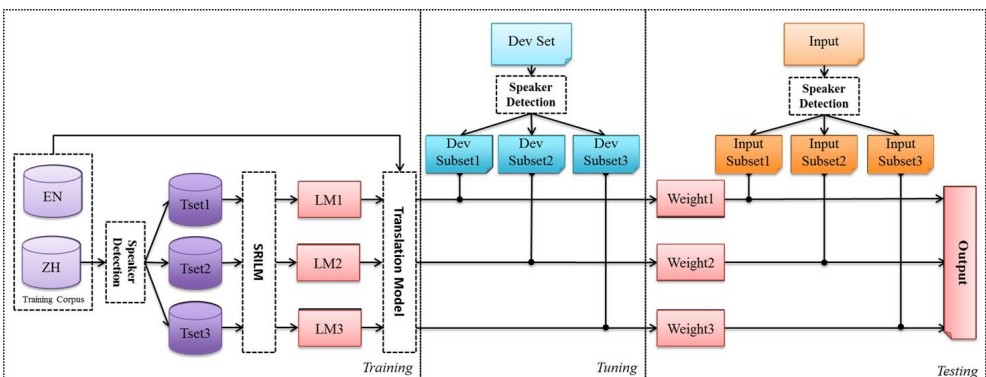

**Figure 8.** Personalized dialogue translation model using speaker information.

### 3.4. Real-Life Applications

Dialogue translation can help real-life systems such as a hotel-booking conversation online system, which can efficiently and accurately assist customers and agents in different languages to reach an agreement in a dialogue for the hotel booking. Ref. [77] showcases a semantics-enhanced task-oriented dialogue translation system with novel features: (1) task-oriented named entity (NE) definition and a hybrid strategy for NE recognition and translation; and (2) a novel grounded semantic method for dialogue understanding and task-order management.

In the hotel booking scenario, customers and agents speak different languages. For instance, the rest of the paper will assume that customers speak English and agents speak Chinese. Customers access the hotel website to request a conversation, and the agent accepts the customer's request to start the conversion. Figure 9 shows the detailed workflow of the hotel-booking translation system. They first recognize entities by inferring their specific types based on information such as contexts, speakers, etc. Then, the recognized entities will be represented as logical expressions or semantic templates using the grounded

semantics module. Finally, candidate translations of semantically represented entities will be marked up and fed into a unified bi-directional translation process.

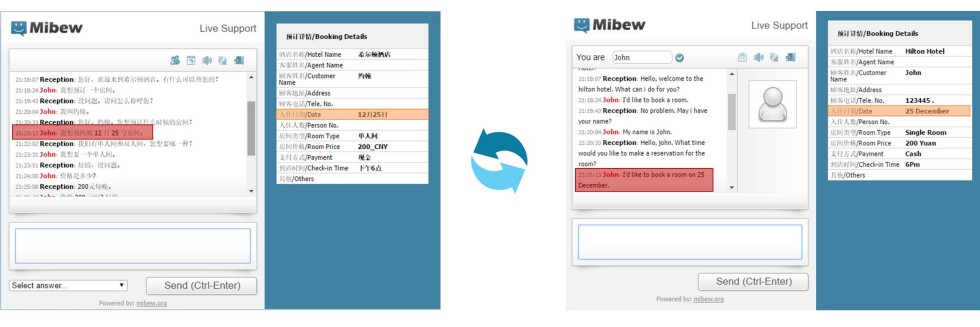

**Figure 9.** Illustration of decoding with reconstruction.

## 4. Building Advanced Dialogue NMT Systems

Prior related work has investigated different inherent characteristics of dialogue translation [11,13–17,63]. In the meantime, a number of advanced techniques have been empirically validated for general-domain translation, which may be adopted to task-oriented translation scenarios, including data selection, back-translation, and larger batch learning. Therefore, we explore a breadth of established approaches for building better dialogue translation systems. First, we mainly investigate three kinds of mainstream models: sentence-level NMT, document-level NMT, and non-autoregressive NMT models [3,13,78]. Technically, we adapt the most recent effective strategies to our models, including back translation [79], data selection [80], domain fine-tuning [81], and large batch learning [82]. To alleviate the low-resource problem, we employ large-scale pre-training language models including monolingual BERT [83], bilingual XLM [84] and multilingual mBART [85], of which knowledge are transferred to translation models. Based on systematic comparisons, we combine the effective approaches to build two SOTA dialogue translation systems w/ and w/o pre-training, respectively.

### 4.1. Methodology

**Sentence-level NMT Models**. We choose the state-of-the-art Transformer network [3] as our model structure, which consists of an encoder with 6 layers and a decoder with 6 layers. For sentence-level NMT (SENT), we use two settings customized from the *base* and *small* configurations. We followed the base configurations to train the SENT-B model, where the dimension of word embedding and the inner feed-forward layer is 512 and 2048 respectively. The parameters of source and target word embeddings and the projection layer before softmax are shared. The number of attention heads is 8. Due to data limitations, we also use the *small* configurations to build SENT-S models. The main differences with the *base* settings are: the inner feed-forward layer is 1024, with the number of attention heads being 4. For all models, we empirically adopt large batch learning [82] (i.e., 4096 tokens $\times$ 8 GPUs vs. 16348 tokens $\times$ 4 GPUs) with a larger dropout of 0.3. The models are trained by the Adam optimizer [86] with $\beta_1 = 0.9$, $\beta_2 = 0.98$. We use the default learning rate schedule used in [3] with the initial learning rate $5 \times 10^{-4}$. Label smoothing [87] is adopted with a value of 0.1. We set the max learning rate to $7 \times 10^{-4}$, warmup steps to 16 K and total training steps to 70 K. All models are trained on NVIDIA V100 GPUs.

**Document-level NMT Models**. For document-level NMT (DOC), we re-implement the cross-sentence model [13] on top of TRANSFORMER-BASE. The addition encoder reads $N = 3$ previous source sentences as history context, and the representations are integrated into the standard NMT for aiding the current sentence. We follow Zhang et al. [88] to use two-stage training, where the context-agnostic model is trained in the first stage (70 K), and then context-aware parameters are tuned in the second stage (40 K).

**Non-autoregressive Models**. Different from autoregressive NMT models that generate each target word conditioned on previously generated ones, Non-autoregressive NMT (Nᴀᴛ) models break the autoregressive factorization and produce target words in parallel [89] as $p(\mathbf{y}|\mathbf{x}) = p_L(T|\mathbf{x};\theta) \prod_{t=1}^{T} p(\mathbf{y}_t|\mathbf{x};\theta)$. Although NAT is proposed to speed up the inference, we expect it can alleviate sequential error accumulation and improve the diversity in conversational translation. We employ the advanced MaskPredict model [78] with a better training method [90]. More specifically, the Mask-Predict uses the conditional mask LM [83] to iteratively generate the target sequence from the masked input. We followed its optimal settings to keep the iteration number as 10 and the length beam as 5. We closely followed previous works to apply sequence-level knowledge distillation to NAT [91]. We train Bɪɢ Transformer as the *AT teachers* and adopt a large batch strategy (i.e., 458 K tokens/batch) to optimize the performance. Traditionally, NAT models are usually trained for 300K steps on regular batch size (i.e., 128 K tokens/batch). In this work, we empirically adopt large batch strategy (i.e., 480 K tokens/batch) to reduce the training steps for NAT (i.e., 70 K). Accordingly, the learning rate warms up to $1 \times 10^{-7}$ for 10 K steps and then decays for 60 K steps with the cosine schedule. For regularization, we tune the dropout rate from [0.1, 0.2, 0.3] based on validation performance in each direction, and apply weight decay with 0.01 and label smoothing with $\epsilon = 0.1$. We use Adam optimizer [86] to train our models. We followed the common practices [78,92] to evaluate the performance on an ensemble of top 5 checkpoints to avoid stochasticity.

**Pre-Training for NMT**. To transfer the general knowledge to downstream tasks, we explore to initialize (part of) parameters of our models with different pre-trained models. In our preliminary experiments, we found that it is difficult for pre-training to improve general-domain NMT models, which usually have a large amount of parallel data. On the contrary, pre-training can help a lot for low-resourced scenarios such as dialogue translation. Furthermore, pre-training on such a large contiguous text corpus enables the model to capture long-range dialogue context information, which motivates us to systematically exploit various kinds of pre-training models in terms of architectures and languages. Ref. [93] shows that large scale generative pre-training could be used to initialize the document-level NMT by concatenating the current sentence and its context. Accordingly, we follow their work to build the Bᴇʀᴛ→Dᴏᴄ model. Besides, ref. [84] proposes directly training a novel cross-lingual pre-training language model (XLM) to facilitate translation tasks. Accordingly, we adopt XLM pre-trained model to sentence-level NMT (Xʟᴍ→Sᴇɴᴛ). More recently, ref. [85] proposes a sequence-to-sequence denoising auto-encoder pre-trained on large-scale monolingual corpora in many languages using the BART objective. We also export mBART for sentence-level NMT (ᴍBᴀʀᴛ→Sᴇɴᴛ).

*4.2. Experiments*

**Setup**. All models are implemented on top of the open-source toolkit Fairseq [94]. Experiments are conducted on two task-oriented translation datasets: WMT20 En-De (http://www.statmt.org/wmt20/chat-task.html, accessed on 20 November 2021), which only consist of 14 K sentence pairs. They contain utterances in task-based dialogues with contextual information, and we use both w/ and w/o context formats for corresponding models. We use the official validation and test datasets for a fair comparison with previous works. Table 2 shows the statistics of WMT20 En-De data. We also use large WMT20 news data (http://www.statmt.org/wmt20/translation-task.html, accessed on 20 November 2021), and select parts of them as pseudo-in-domain data. After preprocessing, we generate subwords via Joint BPE [95] with 32K merge operations. We evaluated the translation quality with BLEU [34].

**Table 2.** Data statistics of En-De after pre-processing. The in-domain/valid/test set is speaker-ignored combined and their average lengths are counted based on En/De.

| Data | # Sent. | # Ave. Len. |
|---|---|---|
| | *Parallel* | |
| In-domain | 13,845 | 10.3/10.1 |
| Valid | 1902 | 10.3/10.2 |
| Test | 2100 | 10.1/10.0 |
| Out-of-domain | 46,074,573 | 23.4/22.4 |
| +filter | 33,293,382 | 24.3/23.6 |
| +select | 1,000,000 | 21.4/20.9 |
| | *Monolingual* | |
| Out-of-domain De | 58,044,806 | 28.0 |
| +filter | 56,508,715 | 27.1 |
| +select | 1,000,000 | 24.2 |
| Out-of-domain En | 34,209,709 | 17.2 |
| +filter | 32,823,301 | 16.6 |
| +select | 1,000,000 | 14.5 |

**Comparison of Advanced Models**. Table 3 illustrates the translation performances of various NMT models with different fine-tuning strategies. As seen, all models are hungry for larger in-domain data due to the data limitation problem (IN+OUT vs. IN). About sentence-level models, the "base + big batch" setting performs better than the "small" one (SENT-B vs. SENT-S). However, it is difficult for document-level models to outperform sentence-level ones (DOC vs. SENT). The interesting finding is that the document-level model trained on pseudo contexts ("IN+OUT") can improve the baseline that is trained on only real context ("IN") by +5.47 BLEU points. There are two main reasons: (1) it lacks large-scale training data with contextual information; (2) it is still unclear how the context help document translation [96,97]. About NAT models, it can improve the vanilla NAT by +0.6 BLEU point, which is lower than those of autoregressive NMT models. About pre-training, we first investigate SENT→DOC. Unfortunately, it is still lower than pure sentence-level models. The performance of BERT→DOC is much better than pure document-level models (56.01 vs. 51.93), which confirms our hypothesis that contextual data is limited in this task. Furthermore, the XLM→SENT can obtain 59.61 BLEU points, which is close to that of SENT-B. Surprisingly, the MBART→SENT with CC25 pre-trained model can achieve the best performance among all models (62.67 BLEU). Except for MBART, all pre-training models cannot beat the best sentence-level model. This demonstrates: (1) it is difficult to transfer general knowledge to downstream tasks; (2) multilingual knowledge may be useful to dialogue scenarios. Encouragingly, we find that the best model with mBART pre-training pushes the state-of-the-art performance on WMT20 English-German dataset up to 62.67 BLEU points.

**Table 3.** Comparison of different models with different fine tuning strategies on De⇒En task.

| Systems | Finetune | BLEU |
|---|---|---|
| | *Models* | |
| Sent-B | In | 42.56 |
| | In+Out | **59.81** |
| Sent-S | In | 41.87 |
| | In+Out | 58.62 |
| Doc | In | 45.65 |
| | In+Out | 51.12 |
| | In→In | 51.93 |
| Nat | In+Out | 54.01 |
| | In+Out | 54.59 |
| | *Pre-training* | |
| Sent→Doc | Out→In | 49.77 |
| | Out→In+Out | 51.58 |
| Xlm→Sent | In+Out | **59.61** |
| Bert→Doc | In+Out | 56.01 |
| mBart→Sent | In+Out | **62.67** |

**Effects of Domain Fine-tuning**. Modeling all the speakers and language directions involved in the conversation can be regarded as a different sub-domain. We conduct domain adaptation for different models to avoid performance corruption caused by domain shifting in Table 4. Specifically, we fine-tune the well-trained models w/ and w/o domain adaptation, denoted as "-Domain" and "+Domain", and evaluated them on domain *combined* and *split* valid sets. As seen, domain adaptation helps a lot on valid set ("Ave." 61.48). While evaluating on combined valid sets has a bias towards models without domain adaptation. We attribute this interesting phenomenon to personality and will explore it in the future.

**Table 4.** Effects of domain adaptation strategy on different De⇒En models. The "AVE." represents averaged score over three models under Split Valid Set.

| Models | −Domain | +Domain |
|---|---|---|
| | *Valid Set (combined)* | |
| Sent-S | **62.66** | 61.19 |
| Sent-B | **64.99** | 63.00 |
| Xlm | 64.19 | **61.30** |
| | *Valid Set (split)* | |
| Sent-S | 60.05 | **62.09** |
| Sent-B | 59.64 | **63.31** |
| Xlm | 61.12 | **62.04** |
| Ave. | 62.27 | **62.48** |

## 5. Conclusions

Dialogue MT is a relatively new but very important research topic to promote MT for practical use. This paper gives the first comprehensive review of the problems, resources, techniques mainly being developed in the last several years. First, we systematically define four critical problems in dialogue translation by reviewing a large number of related works. Second, we collect nearly all existing corpora for dialogue translation task, covering 5 language pairs and 4 sub-domains. Third, we also respectively introduce three representative approaches on architecture, discourse phenomenon and personality aspects. Last, we

discuss an example of real-life applications, demonstrating the importance and feasibility of dialogue translation system. Furthermore, we explore the potential of building a state-of-the-art dialogue translation system by leveraging a breadth of established approaches. Experiments are conducted on a task-oriented translation dataset that is widely used in previous studies (i.e., WMT20 English-German). Encouragingly, we push the SOTA performance up to 62.7 BLEU points on the benchmark by using mBART pre-training method. We hope that this survey paper could significantly promote the research in dialogue MT.

**Author Contributions:** Conceptualization, L.W. and S.L.; methodology, L.W.; software, L.W. and S.L.; validation, L.W. and S.L. and Y.S.; formal analysis, S.L. and Y.S.; investigation, L.W. and S.L. and Y.S.; resources, L.W. and S.L. and Y.S.; data curation, Y.S.; writing—original draft preparation, L.W. and S.L.; writing—review and editing, Y.S. and S.L. All authors have read and agreed to the published version of the manuscript.

**Funding:** Yuqi Sun was supported by the Multi-Year Research Grant (MYRG) funds provided by the University of Macau, grant number MYRG2020-00261-FAH.

**Institutional Review Board Statement:** Not applicable.

**Informed Consent Statement:** Not applicable.

**Data Availability Statement:** Not applicable.

**Conflicts of Interest:** The authors declare no conflict of interest.

## Abbreviations

The following abbreviations are used in this manuscript:

| | |
|---|---|
| NLP | Natural Language Processing |
| MT | Machine Translation |
| NMT | Neural Machine Translation |
| RBMT | Rule-based Machine Translation |
| SMT | Statistical Machine Translation |
| LM | Language Model |
| NE | Named Entity |
| ZP | Zero Pronoun |
| SOTA | State-of-The-Art |

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
