# Peer review of "Recent Advances in Dialogue Machine Translation"

_information, doi:10.3390/info12110484_

Round 1
Reviewer 1 Report
Recent Advances in Dialogue Machine Translation
This paper reviews dialogue translation - an application for machine translation (MT), examining perspectives, resources, approaches, and applications.
The paper is well done in terms of presenting briefly many relevant approaches for this problem area, but does a job less well in informing the reader about many relevant details. As an example in section 3.3.3, the text says: “..baseline MT engine using Moses [71] on MVSub Corpus. They train a 5-gram language model using the SRI Language Toolkit [72] on the target side of parallel corpus. Besides, they use GIZA++ [73] for word alignment and minimum error rate training.” This is indeed a succinct summary of these relevant papers; however, the text explains none of these technical terms; i.e., the reader is left to wonder what are these ideas: baseline MT engine, Moses, MVSub Corpus, 5-gram language model, target side, GIZA++, word alignment and minimum error rate training. They are not explained here or elsewhere in the text.
As the paper does a good job in citing relevant work, one can forgive a certain lack of explanation of useful details, as the reader can readily seek out these citations.
Specific points:
..(i.e. 4 perspectives), .. ->
.. (e.g., 4 perspectives), ..
(The paper improperly uses “i.e.” where “e.g.,” is intended; these are examples; also, add the comma)
… also exploit to building a state-of-the-art .. ->
… also build a state-of-the-art ..
..BLEU points on the commonly-used benchmark by .. ->
..BLEU points on a commonly-used benchmark by ..
..(MT) for breaking language barrier ..
..(MT) for improving communication (or translation) ..
.. improve translation quality of dialogue translation system. .
. improve translation quality of dialogue translation systems.
.. some researchers explored to construct data for modelling ..
.. some researchers have explored ways to construct data for modelling ..
.. only a few corpora can be used for dialogue ..
.. only a few corpora that can be used for the dialogue ..
.. translating dialogue conversations are less ..
.. translating dialogue conversations has been less ..
.. direction investigates to incorporating the dialogue ..
.. direction investigates incorporating dialogue ..
..One the other hand, some researches exploited to explicitly model various .. ->
..On the other hand, some research has explicitly modelled various ..
.. there have been more interests in modelling dialogue machine ..
.. there has been more interest in modelling dialogue machine ..
.. corpora for dialogue translation task, ..
.. corpora for the dialogue translation task, ..
..example of real-life application, ..
..example of real-life applications, ..
.. by using mBART pre-training method.
.. by using the mBART pre-training method.
.. We explore to build a state-of-the-art ..
.. We explore building a state-of-the-art ..
..by combing advanced techniques in Section 4.
..by combining advanced techniques in Section 4.
.. knowledge on translation model and ..
.. knowledge on translation models and ..
.. progress towards to constructing and ..
.. progress towards constructing and ..
.. is a reference-based and computed ..
.. is reference-based and computed ..
.., Figure 2 and 3 show examples of..
.., Figures 2 and 3 show examples of..
.. than single sentence and at least as much than textual discourse.
.. than a single sentence and at least as much as textual discourse.
…3.4) for dialogue machine translation task.
…3.4) for a dialogue machine translation task.
..investigated different approaches MT and evaluation, which ..
..investigated different approaches for MT and evaluation, which ..
… Zero Anaphora is a more complex case .. - describe what this is
..and a number of work exploited to address this problem [16,50–52].
..and several works have addressed this problem [16,50–52].
.. monolingual data which cannot used for MT [5–8].
.. monolingual data which cannot be used for MT [5–8].
..The detailed corpora for dialogue translation task are ..
..The detailed corpora for the dialogue translation task are ..
.. Previous work on dialogue translation usually randomly select some ..
.. Previous works on dialogue translation usually randomly select some ..
.. annotated with zero pronouns .. - does this mean avoiding use of pronouns?
.. as well as zero anaphora .. - as this term is noted more than once, please explain it
.. contains speaker tags and scene boundary which are ..
.. contains speaker tags and scene boundaries, which are ..
.. length (|L|) and related work. - where is this “related work”? The legend describes all columns.
..representations are fed into NMT decoder to ..
..representations are fed into the NMT decoder to ..
.. document-level representation, which is further fed into NMT decoder.
.. document-level representation, which is further fed into the NMT decoder.
..with those of standard NMT encoder.
..with those of the standard NMT encoder.
.. method to annotate ZPs … - what are “ZPs”?
.. such as hotel-booking conversation online system, ..
.. such as a hotel-booking conversation online system, ..
.. [75] showcase a semantics-enhanced ..
.. [75] showcases a semantics-enhanced ..
.. For instance, The rest of the ..
.. For instance, the rest of the ..
..workflow of hotel-booking translation system.
..workflow of the hotel-booking translation system.
(I will cease to note all the omissions of require article, as there are many…)
..About Sentence-level NMT (SENT), We use ..
..About Sentence-level NMT (SENT), we use ..
..conducted on two task-oriented translation dataset: ..
..conducted on two task-oriented translation datasets: ..
.. points which are closed to that of SENT-B.
.. points, which is close to that of SENT-B.
.. models can not beat the best sentence-level model.
.. models cannot beat the best sentence-level model.
.. Findings of the wmt 2020 shared task..
.. Findings of the WMT 2020 shared task..
Author Response
We thank the reviewer for expressing interest and giving insightful comments which will serve to improve the paper considerably. We will attend to all comments to the best extent (all related revisions to the manuscript are marked up using red color).
- Missing some relevant details.
We added a new subsection 2.1.1 to introduce relevant background on statistical machine translation, which provides more details about technical terms (e.g. Moses, n-gram LM, target side, GIZA++, and word alignment) in Section 3.3.3. We will add more explanations in the final version to make the paper much clearer and reader-friendly.
- Missing statements, typos, and grammar errors.
Thanks for pointing out the important missing statements about zero pronouns (ZPs), which have been extensively explained and referred in the new version. We will check all terminologies to ensure that definitions are clear.
All grammatical errors, typos, reference formats have been revised, especially on “i.e.”. In the next version, we will ask a native speaker for proofreading.
In Table 1, we deleted related work due to the limits of the table width.
Reviewer 2 Report
Authors deal with machine translation of dialogs, which would benefit in costs of implementation of dialog system for multilingual purposes. Idea is not new but it is beneficial. For the research itself it is well written and structured however I would like to see highlights in the introduction or in summary. Problematic part for me is evaluation. The datasets authors uses are known to have duplicates and not naturally high BLEU would guide us to a conclusions that training data included duplicates of text set. I recommend to check is toughly and re-do the evaluation. Adding some metrics would also be beneficial. Yes BLEU is most popular but not most reliable.
Author Response
We thank the reviewer for insightful comments which will serve to improve the paper considerably. We will attend to all comments to the best extent (all related revisions to the manuscript are marked up using blue color).
- Missing highlights.
We added highlights of the whole paper in the Introduction as follows:
- Previous works mainly exploited dialogue MT from perspectives of coherence, consistency, and cohesion. Furthermore, recent studies began to pay more attention to the issue of personality such as role preference.
- Although there are some related corpora, the scarcity of training data is one of the crucial issues. This severely hinders the further development of the deep learning methods for real applications of dialogue translation.
- Existing approaches can be categorized into three main strands. One research line is to exploit document-level NMT architectures, which can improve the consistency and coherence in translation output. The second one tries attempts to deal with specific discourse phenomena such as anaphora, which can lead to better cohesion in translations. The third line aims to enhance the personality of dialogue MT systems by leveraging additional information labeled by humans. In future work, it is necessary to design an end-to-end model that can capture various characteristics of dialogues.
- Through our empirical experiments, we gain some interesting findings: 1) data selection methods can significantly improve the baseline model especially for small-scale data; 2) the large batch learning works well, which makes sentence-level NMT models perform the best among different NMT models; 3) document-level contexts are not always useful on the dialogue translation due to the limitation of data; 4) it is helpful to dialogue MT by transferring general knowledge from pre-trained models.
- About duplicates and evaluation.
Sorry, I don't quite understand this viewpoint, but I will try to explain it.
Duplication in the WMT20 chat dataset is not concerned by us, because 1) The corpus is document-level. Even if individual sentences are duplicated, the context is usually different. Our document-level or pre-training models consider contexts instead of single sentences. 2) duplication is useful to low-resource MT models. Thus, one commonly used technique is duplicating data for N times. To make it simple, we did not duplicate or remove duplications in our experiments. In general, duplication is not a problem for MT according to my experience.
The BLEU scores reported in the article is comparable to Table 4 of Findings of the WMT 2020 Shared Task on Chat Translation (https://aclanthology.org/2020.wmt-1.3.pdf). Our mBART pre-trained model achieves 62.7 BLEU points, which has surpassed the best system in the WMT2020 shared task (62.4 BLEU points) on German-to-English. We did not report the TER score because it has the same trends as BLEU. We have checked our evaluation (correct) and will try to human evaluation if the final version (make it more reliable).
We found a typo in the caption of Table 3, which should be German-to-English instead of English-to-German. I wonder if this typo makes the reviewer confused about our results. Sorry about that.
Reviewer 3 Report
The paper presents relevant research on dialogue translation and emphasizes different problems that might arise. The authors collected existing corpora for the dialogue translation task and introduced representative approaches in terms of architecture, discourse phenomenon and personality aspects. The paper also presents a dialogue translation system by leveraging several established approaches.
Suggestions for improvements:
- provide more details on whether you have experienced any bias from pre-training, regardless of fine-tuning
- provide some additional (transformer) hyper-parameters (activation, layers, heads etc.)
- provide additional optimization parameters (optimizer, epoch size etc.)
- You mentioned "Technically, we adapt the most recent effective strategies to our models including back translation, data selection, domain adaptation, batch learning, fine-tuning, and domain adaptation."
However, you should emphasize within your text what refers to what strategy, as this is not clear (e.g. back translation).
E.g. provide more details on where domain adaptation is employed, and how?
- do you have any notion of how much every strategy contributed to the overall BLUE scores?
- proofread paper
Minor fixes:
- fix language errors, such as in
"we also exploit to building a state-of-the-art dialogue NMT system" or
"As a result, there are only a few corpora can be used for dialogue translation task" or
"One research direction investigates to incorporating" or
"some researches exploited to explicitly model various" or
"Last, we discuss" or
"Besides, document-level NMT architectures are proposed to implicitly modelling information across sentences "
- rephrase "we explore to build" and whenever you encounter "explore"
- rephrase sentence "Without loss of generality, we provide the fundamental knowledge on translation model and dialogue translation in this section.
- rephrase "tried attempts"
- rephrase "Finally, they assume 50% speakers as Chinese speakers"
- use consistent writing (e.g. modelling or modeling, dialogs or dialogues)
- fix "In stead"
- rephrase "About Sentence-level NMT (SENT), We use standard TRANSFORMER models"
- rephrase "About Document-level NMT"
- fix missing full stop before "Although NAT is proposed"
Author Response
We thank the reviewer for insightful comments which will serve to improve the paper considerably. We will attend to all comments to the best extent (all related revisions to the manuscript are marked up using brown color).
- Suggestions for improvements.
We have tried to provide more details on pre-training, NMT models, training strategy, domain adaptation with results. We also add references to some technical terms such as back-translation. If needed, we can add a paragraph to show definitions of them.
- Missing statements, typos, and grammar errors.
Thanks for proofreading the paper. All grammatical errors, typos, punctuations have been revised. In the next version, we will ask a native speaker for proofreading.
Round 2
Reviewer 2 Report
accept